# Improving the High-Temperature Gate Bias Instabilities by a Low Thermal Budget Gate-First Process in p-GaN Gate HEMTs

**DOI:** 10.3390/mi14030576

**Published:** 2023-02-28

**Authors:** Catherine Langpoklakpam, An-Chen Liu, Neng-Jie You, Ming-Hsuan Kao, Wen-Hsien Huang, Chang-Hong Shen, Jerry Tzou, Hao-Chung Kuo, Jia-Min Shieh

**Affiliations:** 1Department of Photonics, Institute of Electro-Optical Engineering, College of Electrical and Computer Engineering, National Yang Ming Chiao Tung University, Hsinchu 30010, Taiwan; 2Taiwan Semiconductor Research Institute (TSRI), Hsinchu 30078, Taiwan

**Keywords:** low thermal budget gate first process, ohmic contact, HTGB, gallium nitride, power device

## Abstract

In this study, we report a low ohmic contact resistance process on a 650 V E-mode p-GaN gate HEMT structure. An amorphous silicon (a-Si) assisted layer was inserted in between the ohmic contact and GaN. The fabricated device exhibits a lower contact resistance of about 0.6 Ω-mm after annealing at 550 °C. In addition, the threshold voltage shifting of the device was reduced from −0.85 V to −0.74 V after applying a high gate bias stress at 150 °C for 10^−2^ s. The measured time to failure (TTF) of the device shows that a low thermal budget process can improve the device’s reliability. A 100-fold improvement in HTGB TTF was clearly demonstrated. The study shows a viable method for CMOS-compatible GaN power device fabrication.

## 1. Introduction

GaN is a wide bandgap (3.4 eV) semiconductor material with excellent physical properties, such as a high breakdown field (>3.5 MV/cm), high thermal stability, and high saturation electron velocity (~107 cm/s) [1,2,3]. AlGaN/GaN-based high electron mobility transistors (HEMTs) have proven to be outstanding modern power devices with low on-resistance, low gate charge, and high breakdown voltage. The p-GaN gate GaN HEMTs are ideally suited for use as enhancement-mode (E-mode) transistors because of their excellent threshold voltage (V_TH_) control capabilities and require a positive V_TH_ above 1.5 V to turn on the device [4,5,6]. However, V_TH_ instabilities of p-GaN gate GaN HEMTs have been reported and discussed [7,8,9]. The V_TH_ shift is due to several complex reasons, including acceptor-like traps in p-GaN [10,11], p-GaN gate sidewall leakage [12,13,14], low Schottky barrier heights [10], etc. Therefore, the study of V_TH_ instability of GaN-based HEMTs has been the subject of many research reports [15,16,17,18,19,20]. To ensure the safe operation of the power device, the negative V_TH_ drop must be improved. High-temperature gate bias (HTGB) is one of the most suitable methods for assessing device reliability at high temperatures [21].

In this study, we performed the source/drain (S/D) resistive annealing process at various annealing temperatures to observe the threshold voltage instability associated with thermal budget. Typically, the thermal budget of S/D annealing takes place at the highest process temperature, except for GaN epitaxy. Therefore, the threshold voltage instability of GaN-based HEMTs with respect to the thermal budget of S/D annealing is discussed, as there is no previous study on this topic. Based on the study, we found that the negative threshold voltage drop is strongly related to the annealing temperature. In addition, the new ohmic S/D contact process used not only maintains a low contact resistance (0.6 Ω-mm), but also reduces the annealing temperature to 550 °C. The shift in V_TH_ was reduced to −0.74 V after 10^−2^ s of gate stress at 150 °C. The time to failure (TTF) of the device exhibits an improvement of 100 times, thus indicating the improvement in device reliability. Hence, the study suggests that the instability of the threshold voltage could be improved by employing the low heat budget process.

## 2. Device Structure and Fabrication

The fabrication starts by growing the undoped GaN epilayer on a 6-inch low-resistance Si (111) substrate using a low-pressure organic chemical vapor deposition system (MOCVD). The epilayer growth started with AlN/AlGaN buffer layers, followed by an unintentionally carbon-doped (UID) GaN buffer layer. Following the buffer layers, a 300 nm undoped GaN channel layer, 12 nm Al_0.16_Ga_0.84_N barrier layer, and a 90 nm p-GaN were grown. Mg is used as a p-type dopant for p-GaN layer with a concentration of about 5 × 10^19^ cm^−3^. The Mg dopant was activated in situ with metal-organic chemical vapor deposition (MOCVD) after the epilayers had grown. The wafer bow warp was controlled under ±35 μm. The total epilayer thickness is approximately 4.8 μm.

For device fabrication, the 6-inch Si wafer was diced into 1.2 × 1.2 cm^2^ small pieces. The gate-first device fabrication process was initiated by etching the p-GaN gate using fluorine-based gas andwell etched-stop at the AlGaN barrier layer. A surface roughness of about 1 nm is measured at the AlGaN access region. After etching, the gate metal (TiN) was deposited and patterned on p-GaN corresponding to the AlN/Al_2_O_3_ passivation grown by atomic layer deposition (ALD) [22]. Ar-implantation is used to perform the mesa isolation process. Next, a 200 nm SiNx layer was deposited by plasma-enhanced chemical vapor deposition (PECVD) to serve as the second passivation layer. The S/D region was defined using photolithography and etched accordingly. The top passivation/AlGaN stacking layers were fully removed using dry etching, followed by a 2 nm over-etching in the GaN channel layer. In this study, two types of processes are used: Process A (Sample A-1 to Sample A-4) is without an amorphous-Si (a-Si) assisted layer, while Process B (Sample B-1) has an a-Si assisted layer. For Samples A-1 to A-4, the S/D metal layer stack (Ti/Al/Ti with dimensions 25/500/5 nm) was grown by a lift-off process. For Sample B-1, a 1 nm a-Si-assisted layer was included, which is grown by AMAT CENTURA SYSTEM 5200 PECVD prior to the ohmic metal deposition.

To find an optimal thermal budget for device reliability, Sample A-series are studied under various S/D annealing temperatures ranging from 550 °C to 700 °C. Finally, the fabrication is concluded by growing the gate field plate, S/D metal pads, and SiO_2_ passivation. The length of the p-GaN is 2 μm with a 0.5 μm enclosure of TiN gate metal. The dimensions of L_GS_/L_GD_/L_GFPD_/W_G_ are 3/15/3/100 μm, respectively. Transmission line measurement (TLM) is used to extract the contact resistance of the device. The schematic device structure is shown in Figure 1a, and the device schematic process flow in Figure 1b. The optical 3D surface profiler and the TEM of the fabricated device is shown in Figure 2. A part of the GaN epitaxial layer of the S/D recess is also etched so that the two-dimensional electron gas on the side wall can directly contact the metal structure, as indicated by the orange circle in the Figure 2b. The list of fabricated samples along with the corresponding annealing conditions are tabulated in Table 1.

## 3. Results and Discussion

Figure 3a shows the relationship between contact resistance and annealing temperature for the two different process schemes (Process A and Process B). The highest contact resistance was observed at an annealing temperature of 550 °C (Sample A-1) which might be due to insufficient metal for alloying. As the annealing temperature increases to 600 °C (Sample A-2), the contact resistance was reduced to 1.1 ± 0.1 Ω-mm and gradually increases as the annealing temperature rises. The increase in contact resistance can be attributed to the boiling of Al after 650 °C, as shown in Figure 3b. As seen from the optical microscopy images of Sample A-1 (annealing temperature of 550 °C) and Sample A-3 (annealing temperature of 650 °C), the surface of Sample A-3 became worse due to the boiling of Al on the S/D metal surface at 650 °C. In Sample B-1, where a-Si-assisted layer was inserted at the interface between the S/D metal and GaN, the lowest contact resistance up to 0.6 ± 0.2 Ω-mm with a low annealing temperature (550 °C) was exhibited (Figure 3a). To investigate the interface quality of Sample B-1, the high-resolution transmission electron microscopy (HR-TEM) is used. Figure 3c shows the HR-TEM image of a 1 nm a-Si-assisted layer in between S/D metal and GaN of Sample B-1. Furthermore, to observe the composition of the a-Si-assisted layer, energy-dispersive X-ray spectroscopy (EDS) in TEM was carried out. The image of an a-Si-assisted layer deposited at the interface of Ti/GaN after the EDS mapping of the Si signal by TEM is shown in Figure 3d.

To analyze the improvement in the contact resistance, the EDS line scan from top S/D metal to GaN for Sample A-2 and Sample B-1 was compared. The respective EDS line scan for Sample A-2 and Sample B-1 is shown in Figure 4a,b, respectively. In the alloyed region, Al_3_Ti phase was observed at the top of alloyed S/D metal and the formation of TiN layer is observed at the interface for both samples. However, the thickness of the interfacial TiN of Sample B-1 is more than that of Sample A-2. According to [22], the formation of TiN occurs when Ti reacts with GaN. Many nitrogen vacancies (V_N_) were produced near the interface between Ti and GaN after thermal alloying. The shift in Fermi level to the energy level of nitrogen vacancy towards the Fermi level pinning at 0.5 eV below the edge of the conduction band was reported by Lin et al. [23]. The sharp band bending of GaN gives rise to the probability of electron tunneling, and as a result electrons are being transported from GaN to Ti as free carriers. Thus, from images in Figure 4, we concluded that a thin a-Si assisted layer promised a thicker TiN interfacial layer. In addition, the presence of a thicker Ti layer at the top contributes to the reduction in contact resistance. The excess amount of Ti obstructs the reaction between TiN and Al, leading to lower contact resistance.

Figure 5a,b show the measured transfer I_D_-V_D_ characteristics and off-state leakage current characteristics of fabricated HEMTs. As shown in Figure 5b, the excellent off-state leakage current characteristics were obtained within a high drain voltage ranging up to 800 V with an off-state drain current less than 5 × 10^−8^ A/mm at V_DS_ value of 650 V. The BV curves for both processes show a compatible result which indicates that there is no side effect by inserting an a-Si-assisted layer in the ohmic contact process. The on-resistance of the device can be ascribed as the summation of contact resistance (R_C_), source resistance (R_S_), drain resistance (R_D_), and channel resistance (R_Channel_). The on-resistance (R_on_) of the fabricated device was calculated when V_G_ = 6 V and V_D_ = 1 V from I_D_-V_D_ curves. The measured values of series resistances of fabricated GaN HEMT is tabulated in Table 2. The R_on_ shows an improvement from 14.4 Ω-mm (Sample A-2) to 13.0 Ω-mm (Sample B-1), and the corresponding 2 R_C_ reduces from 2.2 to 1.2 Ω-mm by introducing the a-Si assisted layer. The reduction in R_D_ + R_S_ and R_channel_ in Sample B-1 was from the 2DEG-R_S_ reduction from 530 to 510 ohm/sq.

HTGB measurement was performed to compare the thermal budget effect from S/D ohmic metallization process. Both samples, Sample A-2 and Sample B-1, were kept under stress with a high gate forward bias of 7 V at T = 150 °C for 1 ksec while the Drain bias and substrate bias are kept at 0 V. During the measurement, both devices exhibit a V_TH_ shift in subthreshold characteristics; the shifting of V_TH_ with respect to stress time is shown in Figure 6a. The V_TH_ shift was observed at 10^−2^ s, where Sample B-1 is −0.74 V and Sample A-2 is −0.85 V. Sample B-1 exhibits a lower V_TH_ negative shifting than Sample A-2, which was correlated to the thermal budget consideration. The V_TH_ shifting can be attributed to the hole accumulation in the two-dimensional hole gas (2DHG) and hole depletion across the p-GaN/AlGaN/GaN as a p-i-n diode. In a p-i-n diode structure, the hole supply is limited by either tunneling through or thermionic emission (TE) across the Schottky barrier of gate metal/p-GaN. The uniform hole emission over the reverse-biased Schottky diode leads to an increase in the 2DHG density, and consequently the threshold voltage shifting is corresponded to the 2DHG density [14]. Additionally, the out-diffusion of Mg could be the result of p-GaN thermal activation or S/D ohmic metallization annealing. The Mg acceptor-like states in the AlGaN band structure allows the hole trap-assist-tunneling (TAT) by the trapping/de-trapping process, leading to a negative threshold voltage shifting and a serious gate leakage [24], as shown by the green arrow in Figure 6b. Furthermore, as the gate voltage increased the p-GaN potential, holes were injected across the AlGaN barrier into the GaN layer, leaving behind a partially depleted p-GaN layer. Simultaneously, the hole-electron recombination between 2DHG and 2DEG will occur, leading to a positive threshold shift, which is expected to take place after a long stressing time. Tang et al. reported the generation of GaN band-edge ultraviolet (UV) emission when injected holes recombine with 2DEG electrons in the GaN channel layer [25]. This phenomenon can be considered as the reason for positive threshold voltage shifting at 10^1^ s stress time. The positive shifting of both samples is observed to be about ~90 mV from 10^−2^ s to 10^1^ s stress time.

The correlation between gate current and time to failure (TTF) during HTGB at V_G_ = 10 V for Sample A-2 and Sample B-1 is shown in Figure 7. As seen in the figure, the TTF of Sample A-2 is shorter than Sample B-1; this can be attributed to the higher amount of hole injection from Schottky metal through the p-GaN and GaN channel layer [26]. The result shows that a higher thermal budget leads to hole re-activation during S/D metallization. Thus, a higher annealing temperature reactivates the hole doping concentration. Additionally, the sheet resistance of Sample A-2 is lower than Sample B-1 which also suggests the re-activation under S/D metallization. The out-diffused [Mg] into AlGaN served as an acceptor-like trap as well as a negative charge. The re-activation of out-diffused [Mg] enables the removal of negative charges so that the 2DEG-R_S_ can be reduced by a higher annealing temperature. However, post-implantation annealing results in GaN surface decomposition thus limiting the annealing temperature, which results in insufficient activation [26]. Defect assisted due to Mg diffusion results in redistributions of Mg atoms into the deeper region [27].

Stoffels et al. observed the relationship between TTF and hole doping concentration [28]; a higher hole doping concentration and a higher amount of holes injected into p-GaN results in increased gate leakage and poor TTF. Stoffels et al. also reported that enhancing the barrier height of p-GaN/AlGaN/GaN p-i-n diode improves the TTF degradation [14]. Sample B-1 with a lower thermal budget at S/D metallization was able to reduce the TiN/p-GaN alloying temperature, or hole re-activation. In addition, the TTF is improved by 100 times for the lower thermal budget process than that of the higher thermal budget process sample under HTGB testing.

## 4. Conclusions

In summary, we have demonstrated that low contact resistance can be fabricated by using an a-Si-assisted-process scheme. The implementation of an a-Si-assisted scheme not only enhanced contact resistance but also maintained a low annealing temperature. The threshold voltage instability of the device at a high gate stress is also improved by the low thermal budget process. The lower thermal budget process with an a-Si-assisted layer showed an improvement in V_TH_ shifting from −0.85 V to −0.74 V after a 10^−2^ s gate stress at 150 °C. In addition, the lower thermal budget enhanced the TiN/p-GaN alloying and eliminated hole activation from S/D metallization annealing. Furthermore, the TTF of HTGB at gate bias of 10 V at 150 °C was improved by 100 times, thus achieving a significant improvement in the lifetime of the device.

## Figures and Tables

**Figure 1 micromachines-14-00576-f001:**
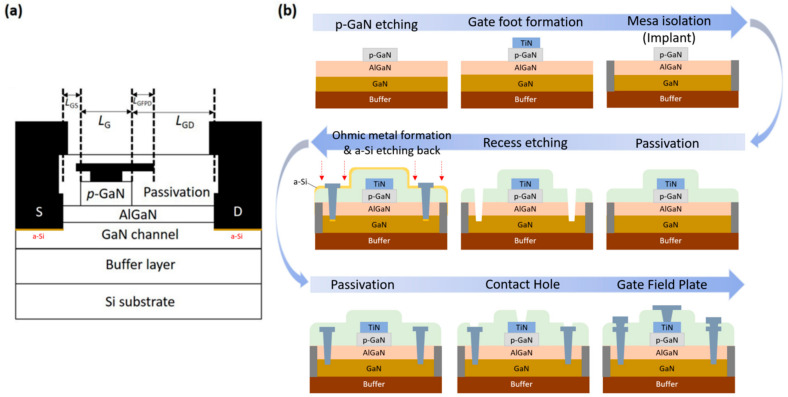
(**a**) Schematic cross-section of the GaN-based HEMT device structure. (**b**) Device process flow.

**Figure 2 micromachines-14-00576-f002:**
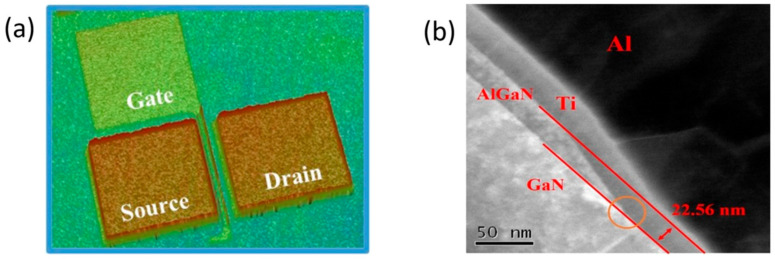
(**a**) Three-dimensional surface profiler. (**b**) TEM image of the fabricated device.

**Figure 3 micromachines-14-00576-f003:**
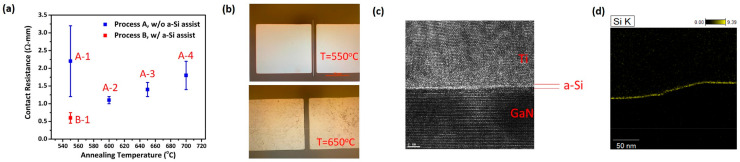
(**a**) Contact resistance as a function of annealing temperature. (**b**) The optical microscopy images of Sample A-1 and Sample A-3, Al boiling can be observed. (**c**) The HR-TEM image of the interface between Ti/GaN corresponds to a 1 nm a-Si-assisted layer of Sample B-1. (**d**) The EDS mapping of Si signal by TEM, the inset shows the corresponding TEM image.

**Figure 4 micromachines-14-00576-f004:**
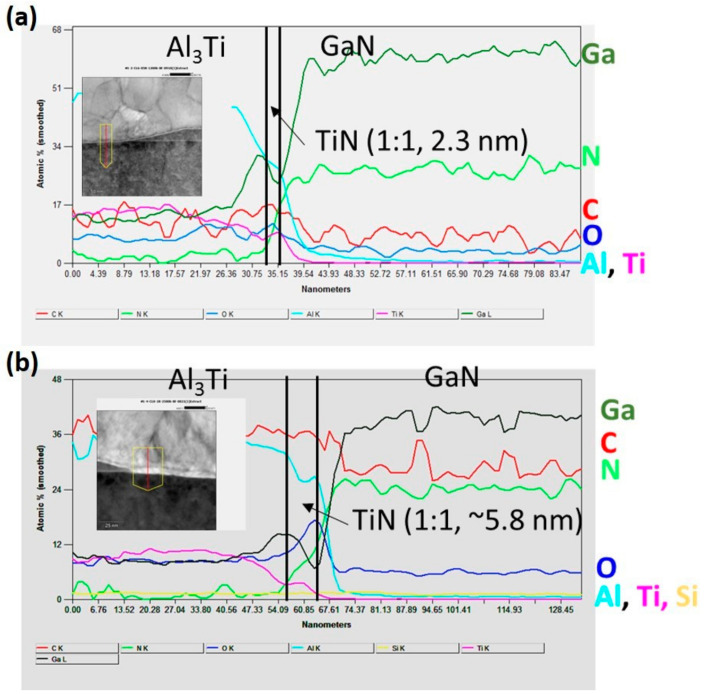
EDS line-scan analysis in the red arrow direction of TEM images for (**a**) Sample A-2 without an a-Si assisted layer and (**b**) Sample B-1 with a 1 nm a-Si-assisted layer. The corresponding TEM images were shown in the inset.

**Figure 5 micromachines-14-00576-f005:**
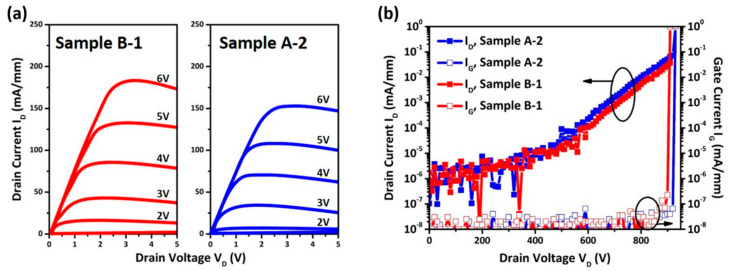
(**a**) The transfer I-V characteristics of Sample B-1 and Sample A-2 under V_D_ = 10. (**b**) Off-state leakage current characteristics.

**Figure 6 micromachines-14-00576-f006:**
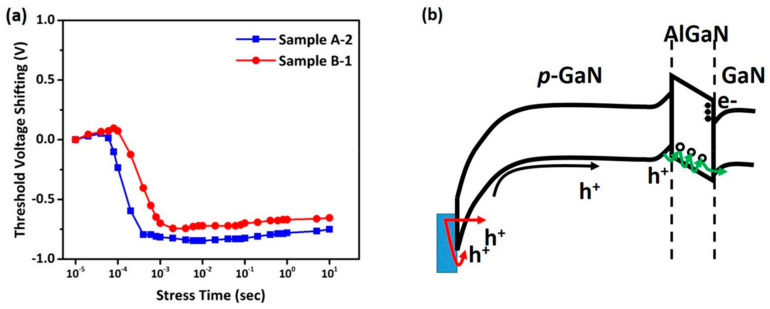
(**a**) Experimental threshold voltage shifting versus accumulated stress time for Sample A-2 and Sample B-1. (**b**) The energy band diagram of the Schottky metal/p-GaN/AlGaN/GaN gate stack at V_G_ = 6 V with different threshold voltage shifting mechanisms. The red arrow indicates the holes across the Schottky barrier height, the green arrow shows the out-diffusion of Mg as an acceptor-like trap level in AlGaN.

**Figure 7 micromachines-14-00576-f007:**
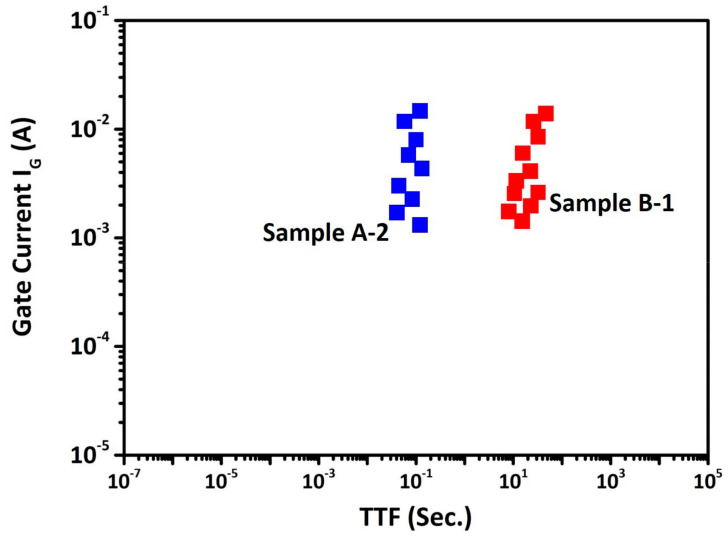
Correlation between gate current and time to failure (TTF) for the device under an HTGB at V_G_ = 10 V, T = 150 °C.

**Table 1 micromachines-14-00576-t001:** Summary of sample list and the corresponding etching depth, a-Si thickness, S/D metal stack, and annealing temperature.

Sample	S/D Etching Depth (nm)	a-Si Thickness (nm)	S/D Metal Stack Ti/Al/Ti (nm)	Annealing Temperature (°C)
A-1	16 + 2	NA	25/500/5	550
A-2		NA	25/500/5	600
A-3		NA	25/500/5	650
A-4		NA	25/500/5	700
B-1		1	50/500/5	550

**Table 2 micromachines-14-00576-t002:** Measured value of series resistances of fabricated GaN HEMT.

Sample	2 R_c_	R_Channel_	R_D_ + R_S_	R_on_
Sample A-2	2.2	1.8	10.4	14.4
Sample B-1	1.2	1.8	10.0	13.0

## Data Availability

The authors confirm that the data supporting the findings of this study are available within the article.

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
