# Peer review of "Improving the High-Temperature Gate Bias Instabilities by a Low Thermal Budget Gate-First Process in p-GaN Gate HEMTs"

_micromachines, 2023, doi:10.3390/mi14030576_

Round 1
Reviewer 1 Report
In this manuscript the authors discuss in detail that using a Si assisted scheme, they can achieve a low contact resistance while maintaining a low annealing temperature in GaN based HEMTs. In this unique study, the authors have discussed the techniques and figures and results in detail. The authors have also demonstrated how this process can lead to improvement in Time to failure or reliability of such devices. However, if the authors can address the following points in their manuscript, I think it wi further help in better understanding of the readers.
1. In Figure 1, the schematic cross section, the authors should also show the Si layer in the bock diagram.
2. Can the process flow be schematically presented for a better understanding?
3. It would be better for the readers to if the authors can provide an optical image of the DUT
4. The g, Wg dimension. How did the authors decide on those dimensions as the gate performance would be significantly affected by this.
5. Positive shift in threshold voltage (lines 180-181). Is it the same both A-2 and B-1?
Reviewer 2 Report
The reviewed manuscript presents the interesting results on gate bias instabilities in p-GaN Gate HEMTs. The paper can be accepted in the Macromachine journal after addressing the following issues:
1. Regarding Fig. 5: how is it possible that an Mg atom can diffuse into AlGaN (green arrow in Fig. 5) after the bias stress? In order to induce the Mg diffusion, a large amount of energy must be delivered. The Mg atoms must overcome the diffusion barrier. Such a process can be induced only after heating samples up to a very high temperature. Therefore, the authors` mechanism of the threshold voltage instability based on the diffusion of Mg into AlGaN is, in my opinion, is invalid. Please correct this issue. 2. The detailed information about the Mg diffusion in GaN the authors can find here: a). T. Kachi et al., Journal of Applied Physics 132, 130901 (2022) https://doi.org/10.1063/5.0107921 b.) M. Matys et al., Appl. Phys. Lett. 121, 203507 (2022); https://doi.org/10.1063/5.0106321 c.) T. Narita et al., Journal of Applied Physics 128, 090901 (2020); https://doi.org/10.1063/5.0022198 The authors should refer to these works about the Mg diffusion.3. What is the reason that the sample B1 exhibited much higher drain current compared to sample A-1 (Fig. 4) . The current collapse is similar for both samples which means that trapping by the surface and interface states is also similar in these cases. Thus there is no physical reason that the sample B1 would exhibit a higher current. Please explain this issue in the revised manuscript.
Round 2
Reviewer 1 Report
I really appreciate the authors for taking the time and addressing the comments. They have modified the figures, texts meticulously and have successfully improved the manuscript by doing so. I think the manuscript can be published in its present form.